# Revisiting CounteRGAN for Counterfactual Explainability of Graphs

**Mario Alfonso Prado-Romero**
Gran Sasso Science Institute
L'Aquila, 67100, Italy
`marioalfonso.prado@gssi.it`

**Bardh Prenkaj**
Sapienza University of Rome
Rome, 00198, Italy
`prenkaj@di.uniroma1.it`

**Giovanni Stilo**
University of L'Aquila
L'Aquila, 67100, Italy
`giovanni.stilo@univaq.it`

## Abstract

Counterfactual explainability (CE) has been widely explored in various domains ranging from medical image diagnosis to self-driving cars. Graph CE (GCE), on the other hand, and especially, generative-based GCE has yet to be explored. Here, we adapt CounteRGAN, an image-based generative approach, to consider graph adjacency matrices as special black-and-white images and sample valid counterfactuals directly from the learnt latent space probabilistic distribution.

## 1 Introduction

Graphs are increasingly used in domains such as social networks, biology, and transportation systems. Deep learning techniques can solve graph-related problems, including community detection (Wu et al., 2022), link prediction (Wei et al., 2022), and session-based recommendations (Xu et al., 2021; Wu et al., 2019). However, they lack interpretability, making them black-box models. Graph counterfactual explanation (GCE) methods answer the question of how to change the graph to achieve a different outcome. Thus, the GCE methods enable domain experts to interpret predicted outcomes by inspecting which characteristics of the new graph induced the change in the outcome w.r.t. the original graph. Here, we present an adaptation of CounteRGAN (Nemirovsky et al., 2022) on graphs using the GRETEL framework (Prado-Romero & Stilo, 2022; Prado-Romero et al., 2023) and compare it to SoTA methods. While we are aware that CounteRGAN does not reach desirable performances w.r.t. other heuristic-based methods, we argue that a not-so-straightforward adaptation of image-based neural networks to the graph domain is beneficial. Moreover, we provide the reader with preliminary insights of generative approaches for GCE which are yet to be explored.

## 2 Related Work

The eXplainable AI (XAI) literature distinguishes between inherently explainable and black-box methods (Guidotti et al., 2018). Black-box methods can be further categorised into factual and counterfactual explanation methods. While many works provide explanations for image/text (Dabkowski & Gal, 2017; Selvaraju et al., 2017; Simonyan et al., 2014), only a few focus on graph classification problems (Abrate & Bonchi, 2021). Majority of these works provide factual explanations (Huang et al., 2022; Luo et al., 2020; Yuan et al., 2021). (Abrate & Bonchi, 2021) use a two-stage heuristic to obtain counterfactual instances. Contrarily, (Numeroso & Bacciu, 2021; Wellawatte et al., 2022) use multi-objective reinforcement learning models to generate counterfactuals for molecules. However, these methods are limited in their applicability to other domains. Finally, only (Ma et al., 2022) use generative models for GCE. They exploit a VAE to generate counterfactuals as probabilistic fully-connected graphs with vertex features and a graph structure similar to the input graph.

## 3 Adapting CounteRGAN for graphs

CounteRGAN uses a GAN with residual connections (Zhang et al., 2020) and a classifier to produce meaningful counterfactuals. It was originally proposed for generating grey-scale image counterfactuals, but we adapted it for graphs by interpreting their adjacency matrix as a black-and-white image. The generator produces residual adjacency matrices, which once added to the input graph, allow the

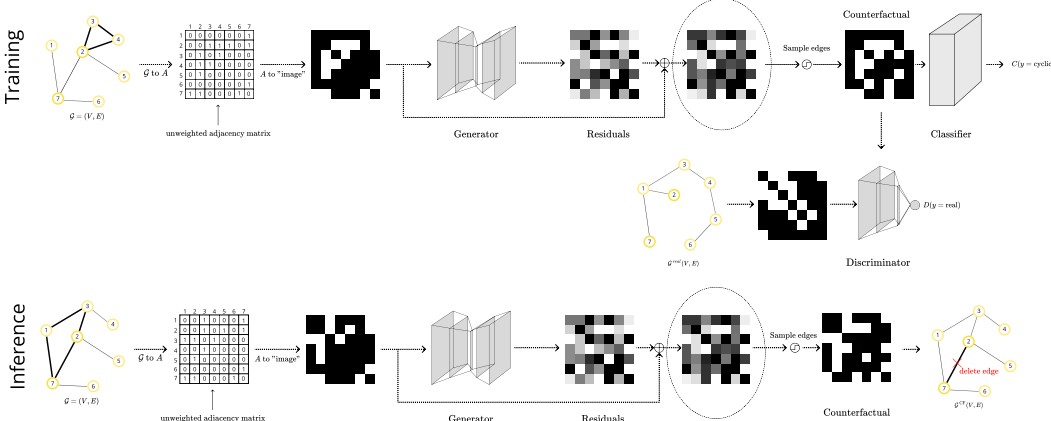

**Figure 1:** CounteRGAN's workflow during training (up) and inference (down). The adjacency matrix is transformed and fed to the generator, which produces counterfactual residuals. The discriminator is trained on both real and generated counterfactuals. At inference time, the generator is used to explain input instances by sampling edges according to learnt probabilistic distribution of its latent space.

additions/removals of the edges. Inspired by the sampling proposed by (Ma et al., 2022), we generate counterfactuals by selecting edges according to the edge probability learned by the generator in the latent space. Furthermore, we train the generator to take as input only the instances of the class we want to explain; whereas, we assign the instances of the opposite class as the real instances of the discriminator. In this way, by forcing the discriminator to distinguish between fake data (the generated instances) and real data (the counterfactual class), the generator will learn to generate counterfactual instances conditioned on the instances of the class to explain. Figure 1 illustrates the train and inference of the adapted CounteRGAN[1].

We assess CounteRGAN's performances using Graph Edit Distance (GED) (Prado-Romero et al., 2023), Correctness (Prado-Romero & Stilo, 2022), Sparsity, Fidelity (Yuan et al., 2022), runtime, and by counting the number of oracle calls. Our results are averaged over 5-fold cross-validations. Note that search-based methods consistently outperform CounteRGAN in all measures. We report in the table captions the number of oracle calls made at training time by CounteRGAN. Contrarily, the number of oracle calls is reported as-is for other search-based and untrained methods.

While CounteRGAN's performances are not satisfactory - notice the difference in terms of GED and correctness with the SoTA methods - we argue that generative counterfactual models possess an enormous potential for future research since one can generate multiple valid counterfactual examples by sampling the generator's latent space (Ma et al., 2022). We believe that more graph-suitable convolution operations (e.g., GCNs) would improve the performances of the overall architecture since they integrate message-passing mechanisms within the neighborhood of a particular vertex and do not view the graph as a flattened structure (i.e., black-and-white adjacency matrix). Further details about the methods, performances and discussion are provided in Appendix A.

## 4 CONCLUSION

We straightforwardly adapted CounteRGAN to the graph domain by considering the adjacency matrix as a special kind of image with pixels "turned on/off". By exploiting the learnt latent space of the generator, we can produce multiple counterfactual explanations by sampling the edge probability encoded within this latent space. This mere adjacency-matrix-to-image adaptation demonstrates that generative approaches have the potential to push the boundaries of the current SoTA. We are aware that sampling on large networks might be a potential issue, and reserve this investigation for future work. Lastly, the integration of graph-based convolution operations is expected to improve the performances of Generative Graph Counterfactual Explainability (GenGCE) methods.

---

[1]The reader can find this implementation at `https://github.com/MarioTheOne/GRETEL`

URM STATEMENT

Authors Mario Alfonso Prado-Romero and Bardh Prenkaj meet the URM criteria of the ICLR Tiny Papers Track.

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

## A  COUNTERGAN'S PERFORMANCE STUDY

**Table 1:** Performances of CounteRGAN and SoTA methods on TreeCycle28. S indicates search-based methods; G stands for generative. The number of oracle calls for CounteRGAN at training time is 264,001.

|   | Explainer | Runtime (s) ↓ | GED ↓ | Num. oracle calls ↓ | Correctness ↑ | Sparsity ↓ | Fidelity ↑ |
|---|-----------|---------------|-------|---------------------|---------------|------------|------------|
|   | DDBS | 7.160 | 63.306 | 1408.173 | 0.558 | 1.136 | 0.558 |
| S | DCE | 0.125 | **42.570** | 501.000 | **1.000** | **0.766** | **1.000** |
|   | OBS | **0.067** | 49.444 | 139.545 | 0.965 | 0.889 | 0.965 |
| G | CounteRGAN | 4.120 | 271.822 | **100.000** | 0.524 | 4.893 | 0.524 |

We compare CounteRGAN with the following search-based methods that are used in the literature for producing graph counterfactual. **Distribution Compliant Explanation (DCE)** is mainly used as an optimistic baseline. It does not make any assumption about the underlying dataset and searches for a counterfactual instance in it. Since it searches for a valid counterfactual of the given input instance, we expect it to reach maximum correctness and fidelity. **Oblivious Bidirectional Search (OBS)** (Abrate & Bonchi, 2021) is a heuristic explanation method that uses a two-stage approach. First, it modifies the original instance until its class changes. Then, it reverts some of the changes while ensuring that the produced counterfactual maintains the new class. **Data-Driven Bidirectional Search (DDBS)** (Abrate & Bonchi, 2021) follows the same logic as OBS. The main difference is that DDBS uses the edges' probability in a certain class of graph to drive the counterfactual search process.

Tables 1, 2, and 3 depict the performances of the methods for three synthetic datasets, respectively, TreeCycle28, TreeCycle32, and TreeCycle48. These datasets contain 500 randomly generated graphs labeled as cyclic (1) or acyclic (0). The suffix in the dataset's name represents the number of vertices for each instance therein.

**Table 2:** Performances of CounteRGAN and SoTA methods on TreeCycle32. S indicates search-based methods; G stands for generative. The number of oracle calls for CounteRGAN at training time is 132,010.

| | Explainer | Runtime (s) ↓ | GED ↓ | Num. oracle calls ↓ | Correctness ↑ | Sparsity ↓ | Fidelity ↑ |
|---|---|---|---|---|---|---|---|
| | DDBS | **0.143** | 73.970 | 1450.520 | 0.538 | 1.161 | 0.538 |
| S | DCE | **0.143** | **50.112** | 501.000 | **1.000** | **0.788** | **1.000** |
| | OBS | **0.143** | 57.542 | 159.260 | 0.964 | 0.905 | 0.964 |
| G | CounteRGAN | 0.298 | 359.698 | **100.000** | 0.504 | 5.659 | 0.504 |

**Table 3:** Performances of CounteRGAN and SoTA methods on TreeCycle48. S indicates search-based methods; G stands for generative. The number of oracle calls for CounteRGAN at training time is 135,001.

| | Explainer | Runtime (s) ↓ | GED ↓ | Num. oracle calls ↓ | Correctness ↑ | Sparsity ↓ | Fidelity ↑ |
|---|---|---|---|---|---|---|---|
| | DDBS | 46.406 | 112.380 | 1415.001 | 0.559 | 1.175 | 0.559 |
| S | DCE | 0.214 | **82.000** | 501.000 | **1.000** | **0.858** | **1.000** |
| | OBS | **0.147** | 89.268 | 237.678 | 0.935 | 0.934 | 0.935 |
| G | CounteRGAN | 6.612 | 1121.550 | **100.000** | 0.506 | 11.737 | 0.506 |

To have a fair comparison of performances across the board, we build a custom classifier for these datasets, which assesses whether the input graph is (a)cyclic and returns 0 or 1. This classifier is a pure oracle that does not require any training but which always guaranteed to produce a correct label (i.e., accuracy is equal to 100%). Note that this classifier is consistently used by all the enlisted methods to induce the generation of valid counterfactuals.

Due to the original (de)convolution operations of CounteRGAN's generator, the input graph $\mathcal{G}(V, E)$ must satisfy $|V| \equiv 0 \pmod 4$ which is a substantial short-coming considering the unconstrained aspects of real-world graphs.

It is interesting to notice the enormous leap - an order of magnitude bigger - in GED[2] for CounteRGAN when passing from graphs with $32$ vertices to $48$. Since CounteRGAN does not support vertex additions/removals, we found out that the model is capable of producing valid counterfactuals for a given acyclic graph in input by simply connecting each vertex with one another (i.e., complete graph); hence, the elevated GED. In contrast, it is incapable of cutting down edges when the input graph is cyclic. Furthermore, due to this shortcoming, the $\sim 0.5$ value for Correctness/Fidelity is justified (i.e., half of the time, CounteRGAN is right at generating a "valid" counterfactual).

---

[2]GED measures the distance between two graphs in terms of edge/vertex additions/removals

