# OpenReview forum: "Revisiting CounteRGAN for Counterfactual Explainability of Graphs"
_ICLR.cc/2023/TinyPapers — Submitted to Tiny Papers @ ICLR 2023_

### Official Review · Reviewer_scGy · 2023-03-31

**Confidence:** 5

**Summary Of Contributions:**

Counterfactual explanations suggest what should be different in the input instance to change the outcome of an ML model. The authors present an adaptation of an image-based generative approach to generate counterfactuals for graphs. Although the proposed method does not improve the SOTA, the authors claim that the approach has potential for further research.

**Rating:**

Great Start (GS): a submission which meets some of the reviewing criteria but has room for improvement

**Strengths And Weaknesses:**

Strength(s):
1. The proposed methodology is clearly illustrated in Figure 1.

Weaknesses:
1. Some key statements need justifications and/or definitions (see suggestions)
2. How does the technique work for massive graphs with millions/billions of nodes? Wouldn't the adjacency matrix image become too large to handle?
3. Does not include any results in the main part of the paper.

**Suggested Changes:**

1. Some statements need justifications and definitions:

	i. Since the paper is about counterfactual explainability, please begin the manuscript with a definition of the same. Readers unfamiliar with, or who are new to the concept will appreciate it. ".. has been widely explored ...", please add a citation or two to back this claim. Consider this as an introduction to the problem that the paper is about.

	ii. There has been a lot of work on explaining the interpretability of deep learning techniques (one example: https://ieeexplore.ieee.org/document/8397411). I would suggest that the authors cite some of these works.

2. Regarding 2. in weaknesses: In case of large graphs, sampling parts of the graph and using the adjacency matrices of them instead may be one alternate approach. Also, there has been some prior work on interpreting the adjacency matrix as an image which may be of interest to the authors. See:

	a. Network classification using adjacency matrix embeddings and deep learning https://ieeexplore.ieee.org/document/7752249/

	b. Network Signatures from Image Representation of Adjacency Matrices: Deep/Transfer Learning for Subgraph Classification https://arxiv.org/pdf/1804.06275

	c. The intrinsic scale of networks is small https://dl.acm.org/doi/pdf/10.1145/3341161.3342893).

---

### Official Review · Reviewer_2Yqt · 2023-03-31

**Confidence:** 5

**Summary Of Contributions:**

This paper adapts CounteRGAN to the graph domain by considering the adjacency matrix as a special kind of image with pixels ”turned on/off”.

**Rating:**

High Impact (HI): a submission which meets the reviewing criteria and is predicted to make an impact on the field

**Strengths And Weaknesses:**

Strengths:
i) The authors have clearly described their proposed approaches.
ii) The paper is well written and organized.
iii) This paper explores the generative-based Graph Counterfactual explainability.

**Suggested Changes:**

No significant change.

---

### Official Review · Reviewer_d1gY · 2023-04-04

**Confidence:** 2

**Summary Of Contributions:**

The paper adapts CounteRGAN and treats graphs adjacency matrices as images to generate counterfactuals. These samples are generated from learnt latent space probabilistic distribution. The paper promises potential in generative counterfactual methodologies. The method does not yield SOTA results but

**Rating:**

Clear, Correct, and Reproducible (CCR): a submission which meets the reviewing criteria

**Strengths And Weaknesses:**

Strength:

The paper is concise and provides clarity on methodology.
The results are analyzed and commentary is attached and future work is identified too.

Weaknesses:
The paper does not provide a clear explanation on why this has an enormous potential

**Suggested Changes:**


Add a description of terms like GED etc for better readability.

---

### Meta-Review · Area_Chair_pxbu · 2023-04-07

**Recommendation:** Invite to archive
**Confidence:** 4

**Metareview:**

This well-written paper meets the Clear, Correct, and Reproducible (CCR) criteria, as agreed upon among the reviewers. It does not, however, provide convincing evidence in the main text on the superiority of the proposed method.

**Summary:**

The paper treats the adjacency matrix as a special black-and-white image and applies CounteRGAN for the generation of graph structures.

**Reason For Not Giving A Higher Recommendation:**

- no evidence, numerical or theoretical, is presented in the main text
- I second reviewer `scGy`'s concern on the feasibility on large graphs
- have you considered the permutation equivariance of graphs?

**Reason For Not Giving A Lower Recommendation:**

- the paper is well written
- the concept is illustrated clearly

---

### Decision · Program_Chairs · 2023-04-09

Invite to archive

---

> ### Author Response · Authors · 2023-05-26
> **Response to meta-reviewer and referees**
>
> We thank the meta-reviewer and the referees for the thoughtful comments. We considered all of them, and we provided a camera version that includes the needed clarifications.